# Combination Therapy versus Monotherapy in the Treatment of *Stenotrophomonas maltophilia* Infections: A Systematic Review and Meta-Analysis

**DOI:** 10.3390/antibiotics11121788

**Published:** 2022-12-09

**Authors:** Abhisit Prawang, Naphatsawan Chanjamlong, Woranattha Rungwara, Wichai Santimaleeworagun, Taniya Paiboonvong, Thidarat Manapattanasatein, Prompiriya Pitirattanaworranat, Pongsakorn Kitseree, Sukrit Kanchanasurakit

**Affiliations:** 1Department of Pharmacy Practice, College of Pharmacy, Rangsit University, Pathum Thani 12000, Thailand; 2Department of Pharmacy, Faculty of Pharmacy, Silpakorn University, Muang, Nakorn Pathom 73000, Thailand; 3Division of Clinical Pharmacy, Department of Pharmaceutical Care, School of Pharmaceutical Sciences, University of Phayao, Phayao 56000, Thailand; 4Center of Health Outcomes Research and Therapeutic Safety (Cohorts), School of Pharmaceutical Sciences, University of Phayao, Phayao 56000, Thailand; 5Unit of Excellence on Clinical Outcomes Research and IntegratioN (UNICORN), School of Pharmaceutical Sciences, University of Phayao, Phayao 56000, Thailand; 6Division of Pharmaceutical Care, Department of Pharmacy, Phrae Hospital, Phrae 54000, Thailand

**Keywords:** *Stenotrophomonas maltophilia*, monotherapy, combination therapy, mortality

## Abstract

*Stenotrophomonas maltophilia* is a multidrug-resistant bacterium that is difficult to treat in hospitals worldwide, leading to high mortality. Published data describing the use of monotherapy or combination therapy and which one is better is still unclear. We aimed to investigate the efficacy of monotherapy and combination therapy in the treatment of *S. maltophilia* infections. We performed a systematic review of combination therapy and additionally a systematic review and meta-analysis to determine the effects of monotherapy versus combination therapy on mortality in *S. maltophilia* infections. Electronic databases: Cochrane Library, PubMed, Embase, ClinicalTrials.gov, Scopus, and OpenGrey were accessed. Of the 5030 articles identified, 17 studies were included for a systematic review of combination therapy, of which 4 cohort studies were finally included for meta-analysis. We found there is a trend of favorable outcomes with respect to mortality in the use of combination therapy to treat complex or severe *S. maltopholia* infections. A meta-analysis of monotherapy showed a statistical significance in the decreasing rate of mortality in hospital-acquired pneumonia (hazard ratio 1.42; 95% confidence interval, 1.04–1.94) compared to combination therapy, but not significant in bacteremia (hazard ratio 0.76; 95% confidence interval, 0.18–3.18). Further studies should continue to explore this association.

## 1. Introduction

*Stenotrophomonas maltophilia* is an aerobic Gram-negative bacterium that can cause various opportunistic infections in humans [1]. The prevalence of *S. maltophilia* infections in Asia, Europe, and Latin America as reported in worldwide surveillance and multi-center studies are 1.68%, 1.0%, and 0.8%, respectively [2]. Importantly, multi-drug resistant (MDR) *S. maltophilia* in hospitals worldwide is emerging, leading to high mortality rates [3]. *S. maltophilia* is intrinsically resistant to various classes of antibiotics, including beta-lactam agents and aminoglycosides [4,5]. The main mechanism of resistance is the presence of genes that encode efflux pumps and antibiotic inactivating enzymes [4]. Consequently, *S. maltophilia* infections are extremely difficult to treat [2] and treatment for those who have a *S. maltophilia* infection, which is always resistant to first-line therapy, or an MDR strain which is resistant to trimethoprim-sulfamethoxazole (TMP-SMX), levofloxacin, amikacin, colistin, and tigecycline [6], is usually various combinations of antimicrobial agents [7]. Regimens using antimicrobial agents have been surveyed extensively in order to achieve an efficient combination that will overcome bacterial resistance and attain synergism when possible. A combination of two to three conventional agents on *S. maltophilia* organisms that are particularly susceptible to these agents often leads to advantageous results, as do new antibiotics such as televancin that have demonstrated synergistic effects on *S. maltophilia* [8].

Trimethoprim-sulfamethoxazole and fluoroquinolones are typically used in the treatment of *S. maltophilia* infections. Finding from the subgroup analysis of a study evaluating the outcomes of *S. maltophilia* bacteremia showed no statistically significant differences between fluoroquinolones and TMP-SMX [9]. However, *S. maltophilia* isolates resistant to TMP-SMX and fluoroquinolones have been reported [6]. Treatment with a combination of two or three antimicrobials to overcome resistance may be an attractive option. Whilst several studies have reported the role of synergistic combination therapy in the management of this difficult to treat infection [8], the efficacy of combination therapy remains incompletely defined [10,11,12] [. Moreover, in clinical practice, it is important to consider both the advantage of synergist effects against bacteria and the disadvantage of additive adverse events of drugs [8]. These studies have still been limited. Therefore, our purpose is to investigate the clinical outcome of combination therapy and the effect of monotherapy versus combination therapy for *S. maltophilia* infections and whether or not these very antithetical approaches affect mortality outcomes.

## 2. Materials and Methods

We performed a systematic review of studies evaluating the efficacy of combination therapy against *S*. *maltophilia* infections. Where possible, a meta-analysis was undertaken to determine the mortality in patients receiving monotherapy or combination therapy for the treatment of *S*. *maltophilia* infections. This study was performed in accordance with the guidance set up by the Preferred Reporting Items for Systematic Reviews and Meta-Analyses (PRISMA) guidelines and was registered with the trial registration number ID: CRD42020210843 under the international prospective register of systematic reviews (PROSPERO: www.crd.york.ac.uk/PROSPERO, accessed on 25 July 2022).

### 2.1. Data Sources and Search Strategy

Articles included in this systematic review and meta-analysis were derived from the following electronic databases: The Cochrane Library, PubMed, Embase, Scopus, ClinicalTrials.gov, and OpenGrey. All articles reporting the use of monotherapy or combination therapy in the management of *S. Maltophilia* infections from the inception of the databases to 25 July 2022 were screened for inclusion. Articles meeting the inclusion criteria were examined. Medical subject headings (MeSH) were applied to each one as applicable. Reference lists of related articles were also explored. The search strategy was carried out with the following keywords: “*Stenotrophomonas maltophilia*”, “mortality”, “therapeutics”, and “anti-bacterial agents” with slight adjustments made as suitable to each database. There was no study design and no language restrictions.

### 2.2. Study Selection

The inclusion criteria included human studies: (1) *S. maltophilia* infections in adults ≥18 years of age, and (2) presented outcomes as odds ratios (OR), risk ratios (RR), or hazard ratios (HR) with a 95% confidence interval (95% CI) or *p*-value or mortality rate, and (3) describing the use of combination therapy and monotherapy in the management of *S. matophilia* infections. Non-human studies, reviews, commentaries, editorials, expert opinions, surveys, letters, conference meeting abstracts, systematic reviews, and meta-analyses were excluded.

### 2.3. Data Extraction and Quality Assessment

Two independent authors examined the search results according to the study selection criteria. Details of each study were extracted and tabulated, including study design, patient population, co-morbidity, severity, reported mortality, type of infection, treatment details, percentage of polymicrobial infections, method of bacterial identification, and funding source. We contacted correspondence authors of the relevant articles for missing information. If the correspondence authors did not respond within a month, the article was excluded. The Risk Of Bias In Non-randomized Studies of Interventions (ROBINS-I) assessment tool was used for quality evaluation of the observational studies. Case reports and case series were evaluated using previously published criteria [13].

### 2.4. Definition and Outcome Measures

The term combination therapy denotes the use of two or more antibiotics and the term monotherapy denotes the use of one antibiotic. The primary outcome was 30-day mortality in patients receiving monotherapy or combination therapy for the treatment of *S. maltophilia* infections. The term “in-hospital mortality” was defined as the number of patients who died during hospital admission.

### 2.5. Statistical Analysis

The outcomes of each study, including OR or RR, were transformed into HR for final analysis. The transformations were made using the following equations [14]:(1)$$\mathrm{RR}=\frac{\mathrm{OR}}{\left(1-r\right)+\left(r*\mathrm{OR}\right)}$$
and
(2)$$\mathrm{HR}=\frac{\mathrm{In}\left(1-\left(\mathrm{RR}*r\right)\right)}{\mathrm{In}\left(1-r\right)}$$
where “r” is the mortality rate from infection causes of the reference group (i.e., those with *S. maltophilia* infections who were treated with monotherapy).

The pooled HR and 95% CIs were calculated using a random-effects model (Mantel–Haenszel method). Statistical heterogeneity between studies was assessed using a *χ*^2^ test of heterogeneity (*p* < 0.10 was defined as indicating significant heterogeneity). The degree of heterogeneity was evaluated using the *I*^2^ statistics, whereby 0%–25% indicated low heterogeneity, 25%–50% indicated moderate heterogeneity, 50%–75% indicated substantial heterogeneity, and 75%–100% indicated considerable heterogeneity. Publication bias was assessed using the funnel plot method and Egger’s test. Review Manager for Windows, version 5.3 (The Cochrane Collaboration, The Nordic Cochrane Centre, Copenhagen, Denmark) was used for meta-analysis and R-3.3.1 for Windows (RStudio, Boston, MA, USA) was used for Egger’s test.

### 2.6. Subgroup and Sensitivity Analysis

We performed subgroup analysis adjusting for the following variables: model of meta-analysis, age, day on ventilator, length of stay in Intensive Care Unit (ICU), hospital length of stay, immunocompromised status, and severity status. Immunocompromised population was defined as patients who have a reduced ability to fight infections and other diseases due to underlying medical conditions such as AIDS, cancer, diabetes, malnutrition, and certain genetic disorders or use of certain medicines or treatments including anticancer drugs, radiation therapy, and stem cell or organ transplant. The severity status was defined as APACHE II score ≥ 16, SOFA score ≥ 2, or Pitt score ≥ 4.

## 3. Results

### 3.1. Search Results and Included Study Characteristics

The full details of our search and study selection process are presented in Appendix A and Figure 1, respectively. Initial search identified a total of 5030 articles. After removing duplicates, 28 articles were retrieved. Of the 28 articles, 11 were excluded due to inappropriate comparisons (*n* = 4), in vitro studies (*n* = 5), editorial (*n* = 1), and no response from author for incomplete data (*n* = 1). A total of 17 studies, published between 1996 and 2022, comprised of case reports (*n* = 9), case series (*n* = 4), and cohort studies (*n* = 4) [10,11,12,15,16,17,18,19,20,21,22,23,24,25,26,27,28] were included in the review. Details of the included studies’ case reports/case series are summarized in Table 1 and Appendix A, and four cohort studies [25,26,27,28] are summarized in Table 2 and Appendix A. Meta-analyses comparing the mortality rate of monotherapy versus combination therapy in the treatment of *S. maltophilia* infections were performed in the four cohort studies [25,26,27,28]. These studies were conducted in the USA, France, and Japan and included those with hospital-acquired pneumonia and bacteremia (a total of 851 individual patients).

### 3.2. Quality Assessment

We considered the overall results to be at moderate to serious risk of bias (Appendix A). The quality assessment of case reports and case series are shown in Appendix A. All studies were assessed for exposure, outcome, follow-up period, and clinical application.

### 3.3. Mortality

The finding of our systematic review suggested that the outcomes of *S. maltopholia* infections in patients receiving combination therapy appear to be clinically favorable. Notably, the survival rates of patients with complex or severe infections such as peritonitis, meningitis, ventilator-associated pneumonia, infective endocarditis, and bacteremia were reported to be 100%, 100%, 100%, 70%, and 70.59%, respectively. However, hemorrhagic pneumonia caused by *S. maltopholia* infection had a 100% death rate, despite treatment with combination antimicrobial therapy. The details are shown in Table 1 and Appendix A.

Results from the random effects model meta-analysis of the four cohort studies suggested that the overall effect on mortality was in favor of monotherapy (HR 1.42, 95% CI 1.04–1.94, *I*^2^ 0%) in the management of *S. maltophilia* hospital-acquired pneumonia. However, we observed no statistical difference in mortality in patients with *S. maltophilia* bacteremia (HR 0.76, 95% CI 0.18–3.18, *I*^2^ 0%). There was no indication of any publication bias in the Egger’s test or Begg’s test (Egger’s test *p*-value = 0.462 and Begg’s test *p*-value = 0.846) (Appendix A). The four studies included for analysis comprised two pneumonia and two bacteremia studies with three and two clinical outcome related mortalities, respectively. The details are shown in Table 2 and Appendix A. Monotherapy was shown to have statistically significant effects on the decreased risk of mortality in *S. maltophilia* hospital-acquired pneumonia (hazard ratio is 1.42; and a 95% confidence interval of 1.04–1.94) with no heterogeneity. We found no evidence of difference in mortality between monotherapy and combination therapy with substantial heterogeneity (*I*^2^ was 55%), as shown in Figure 2. In addition, the first included meta-analysis study prescribed the main antimicrobial therapies to treat *S. maltophilia* infections as TMP-SMX, ciprofloxacin, and levofloxacin as monotherapy, while the following agents were prescribing as combination therapy: TMP-SMX + levofloxacin, TMP-SMX + ciprofloxacin, TMP-SMX + moxifloxacin, TMP-SMX + minocycline, TMP-SMX + ceftazidime, levofloxacin + minocycline, levofloxacin + ceftazidime, ciprofloxacin + minocycline, ciprofloxacin + ceftazidime, and minocycline + ceftazidime. The proportion of the immunocompromised population in this study was 19.8% [25]. The second included meta-analysis study prescribed the main antimicrobial therapies to treat *S. maltophilia* infections as TMP-SMX, ciprofloxacin, and ticarcillin-clavulanate monotherapy and combined in vitro active agents (two or more agents) as combination therapy. The proportion of the immunocompromised population in this study was 37.4% [26]. The third included study in the meta-analysis prescribed TMP-SMX as monotherapy and TMP-SMX + fluoroquinolone as combination therapy. The proportion of the immunocompromised population in this study was 100% [27]. The last included study prescribed TMP-SMX, third-generation cephalosporin, or extended-spectrum penicillin as monotherapy and a combination of these monotherapy agents as combination therapy. The proportion of the immunocompromised population in this study was 97.80% [28]. The details are shown in Table 2.

### 3.4. Subgroup and Sensitivity Analysis

We performed subgroup analysis to explore other factors that might influence mortality associated with *S. maltophilia* infections. The models of meta-analysis, age, duration of ventilator use, length of ICU stay, length of hospital stay, immunocompromised status, and the severity status of the infection were analyzed. The data showed that age < 65 years, duration of ventilator use ≥ 14 days, ICU length of stay < 1 month, hospital length of stay < 1 month, no immunocompromised status, and severe illness favored monotherapy over combination therapy; see Figure 3.

## 4. Discussion

Since the 1980s, monotherapy for *S. maltophilia* infection has involved the use of antimicrobials such as fluoroquinolones or TMP-SMX [9]. To date, the use of a regimen of treatment differs on whether combination or monotherapy is more effective. This study is the first systematic review and meta-analysis evaluating the optimal regimen to treat *S. maltophilia* infections. Four studies were designed to compare the clinical efficacy of both monotherapy and combination therapy, showing that monotherapy was significantly better than combination therapy in terms of mortality outcome in the patients who were infected with hospital-acquired pneumonia and susceptible to antimicrobials. However, the patients who had been infected with *S. maltophilia* bacteremia showed no significant difference with respect to mortality between combination therapy and monotherapy. Our findings suggest that management of *S. maltophilia* hospital-acquired pneumonia or bacteremia should probably be started with monotherapy. Combination therapy should be considered in severe infections or when patients’ conditions do not improve following monotherapy, or if they are infected with an MDR strain. Additionally, the use of novel antimicrobials acting as efflux pump inhibitors might help to treat *S. maltophilia* infections [4].

These suggestions are similar to the recommendation of the Infectious Diseases Society of America Guidance on the Treatment of AmpC β-lactamase-Producing Enterobacterales, Carbapenem-Resistant *Acinetobacter baumannii*, and *S. maltophilia* Infections [29].

Subgroup analysis showed that the use of combination therapy was associated with higher mortality rates than monotherapy in patients with the following factors: age < 65 years, ICU length of stay < 1 month, hospital length of stay < 1 month, no immunocompromised status, severe illness, and duration of ventilator use ≥ 14 days. Although this observation may be potentially due to the additive toxicity secondary to combination therapy and severity of infections, the use of combination therapy may be required in the treatment of infections refractory to monotherapy. Notably, favorable outcomes following use of combination therapy have been reported in patients with severe or complex infections such as bacteremia, infective endocarditis, ventilator-associated pneumonia, and meningitis.

One limitation of our current review is the small number of studies included in the meta-analysis, despite a comprehensive search of electronic database and grey literatures. Although randomized controlled trials would be the gold standard to evaluate the efficacy of treatment against *S. maltophilia* infections, data from observational studies may provide some insights as to the outcome of this rare infection. Notably, results from the Egger’s test and Begg’s test suggested minimal publication bias in our finding. A further longitudinal study should explore the optimal treatment regimen for *S. maltopholia* infections.

In conclusion, finding from this systematic review and meta-analysis suggest that there may be a potential role for combination therapy in the treatment of complex or severe cases of *S. maltopholia* infections. Additionally, compared to combination therapy, monotherapy was associated with more favorable outcomes in the management of hospital-acquired *S. maltopholia* pneumonia. A longitudinal study that further explores this association is warranted.

## Figures and Tables

**Figure 1 antibiotics-11-01788-f001:**
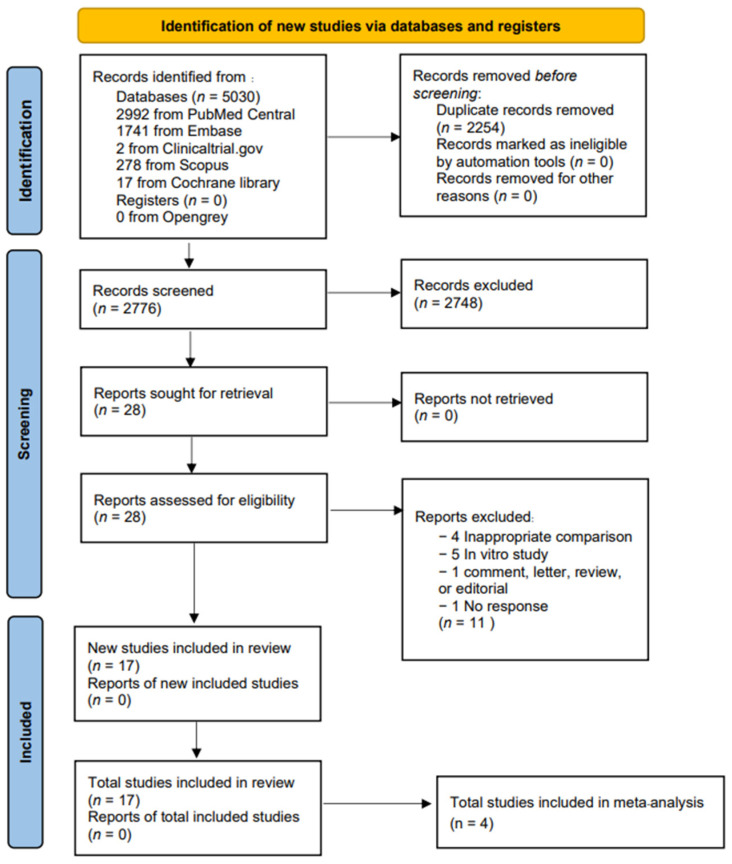
Preferred Reporting Items for Systematic Reviews and Meta-Analyses (PRISMA) flow diagram; summarizes the study selection process.

**Figure 2 antibiotics-11-01788-f002:**
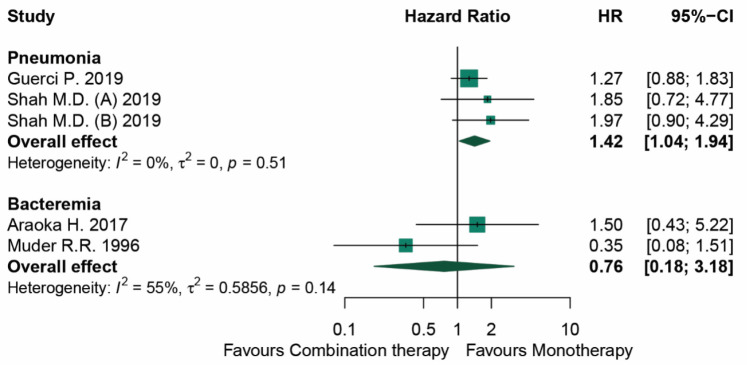
Forest plot presenting the HRs of mortality of patients with *S. maltophilia* infections compared between combination therapy and monotherapy [25,26,27,28].

**Figure 3 antibiotics-11-01788-f003:**
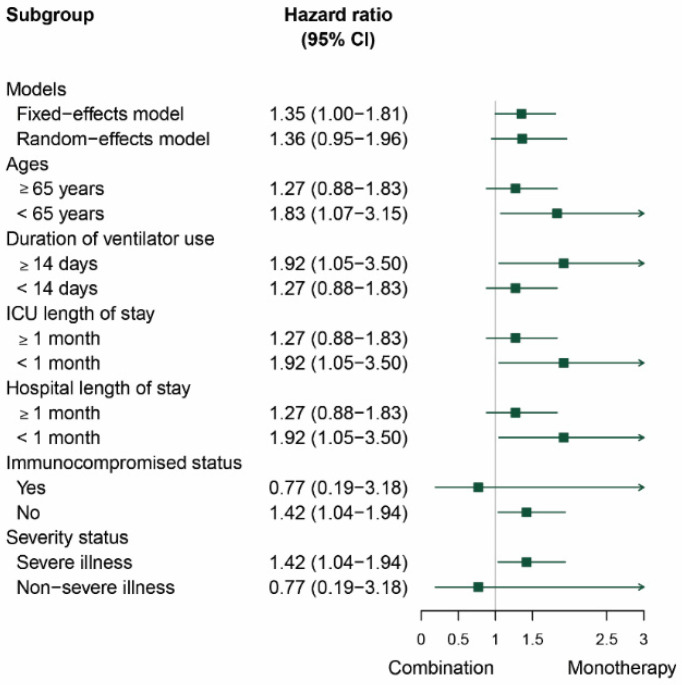
Forest plot presenting the HRs of subgroup analysis of patients with *S. maltophilia* infections compared between combination therapy and monotherapy.

**Table 1 antibiotics-11-01788-t001:** Study characteristics of included studies of combination treatment of *S. maltophilia* infection.

Author (Year)	*Region*	*Design*	*Sample Size*	*Infection*	*Treatment Duration* *(Days)*	*Follow Up Time (Days) (n)*	*Outcome*
*Robert (1996) [28]*	USA	Cohort	18	Bacteremia	N/A	N/A	Died 18%
*Munter (1998) [15]*	Isreael	Case report	1	Infective endocarditis	40	N/A	Died
*Kim (2001) [16]*	Korea	Case report	1	Infective endocarditis	42	60	Clinical response *
*Wood (2010) [17]*	USA	Case report	1	VAP	14	33	Clinical response * Microbiological response **
*Holifield (2011) [18]*	USA	Case report	1	Keratitis	10	N/A	Clinical response * Microbiological response **
*Mori (2014) [19]*	Japan	Case series	8	Hemorrhagic pneumonia	1–16	N/A	Died 100%
*Reynaud (2015) [20]*	France	Case report	1	Infective endocarditis	2	N/A	Died
*Mojica (2016) [21]*	USA	Case report	1	Bacteremia	48	N/A	Microbiological response **
*Subhani (2016) [12]*	India	Case series	28	Infective endocarditis	42 (1), N/A (27)	N/A (27), 60 (1)	Cured 67.86% Died 28.57% N/A 3.57%
*Araoka (2017) [27]*	Japan	Cohort	14	Bacteremia	N/A	30	Died 50%
*Kaito (2018) [22]*	Japan	Case report	1	Bacteremia, Pneumonia	18	N/A	Died
*Payen (2019) [11]*	France	Case series	4	Peritonitis VAP	14	176	Clinical response * 100% Microbiological response ** 100%
*Shah (2019) [25]*	USA	Cohort	38	Pneumonia	N/A	N/A	Died 39.47%
*Guerci (2019) [26]*	France	Cohort	167	Pneumonia	7	N/A	Died 37.72%
*Andrei (2020) [23]*	Romania	Case report	1	Severe pneumonia with pulmonary hemorrhage	7	300	Clinical response * Microbiological response **
*Khanum (2020) [10]*	Pakistan	Case series	2	Meningitis	21	N/A	Clinical response * 100% CSF culture negative 100%
*Petca (2022) [24]*	Romaniav	Case report	1	Pyelonephritis	14	N/A	Microbiological response **

Abbreviations: ED, eye drop; F, female; I, intermediate; M, male; N/A, not applicable; NB, nebulize; R, resistant; S, susceptible; AMC, ampicillin; AMK, amikacin; CAR, carbenicillin; CAZ, ceftazidime; CHL, chloramphenicol; CIP, ciprofloxacin; COL, colistin; FEP, cefepime; GEN, gentamicin; KAN, kanamycin; LVX, levofloxacin; MOX, moxalactam; PEN, penicillin; POL, polymyxin; STR, streptomycin; TIC, ticarcillin; TIM, ticarcillin/clavulanic acid; TMP-SMX, trimethoprim-sulfamethoxazole; TZP, piperacillin/tazobactam; VAN, vancomycin. * Improvement of signs or symptoms of infection related to treatment. ** Generally related to total or partial eradication of isolated organisms.

**Table 2 antibiotics-11-01788-t002:** Characteristics of cohort studies included in the systematic review and meta-analysis.

Author (Year)	*Region*	*Baseline Characteristics*					*Details of Antimicrobials*		*Effect Size* *(95% CI)*	*Severity Score*
		Sample Size	Age of Exposure Group (Year)	Type of Infection	Immuno-Compromised Population (%)	Male (%)	Monotherapy	Combination Therapy		
*Muder (1996) [28]*	USA	91	N/A	Bacteremia	97.8	N/A	TMP-SMX Third-generation cephalosporin Extended-spectrum penicillin	Receiving more than 1 of monotherapy agents	0.35 (0.08–3.18)	Severity score
*Araoka (2017) [27]*	Japan	20	60.5 ^a^	Bacteremia	100	85.71	TMP-SMX	TMP-SMX + fluoroquinolone	1.5 (0.43–5.22)	Pitt score
*Guerci (2019) [26]*	France	282	65 (±9) ^b^	Pneumonia	37.4	69.9	TMP-SMX Levofloxacin Ciprofloxacin Ticarcillin/clavulanate Ceftazidime Minocycline Colistin Rifampicin Tigecycline	N/A	1.27 (0.88–1.83)	SOFA score
*Shah (2019) [25]*	USA	252	62 ^a^	Pneumonia	19.8	62.3	TMP-SMX Levofloxacin Ciprofloxacin Moxifloxacin Minocycline Ceftazidime	TMP-SMX + Levofloxacin TMP-SMX + Ciprofloxacin TMP-SMX + Moxifloxacin TMP-SMX + Minocycline TMP-SMX + Ceftazidime Levofloxacin + Minocycline Levofloxacin + Ceftazidime Ciprofloxacin + Minocycline Ciprofloxacin + Ceftazidime Minocycline + Ceftazidime	(A) = 1.85 (0.75–4.98) (B) = 1.97 (0.96–4.55)	APACHE II score

a, mean age (year); b, mean (SD); N/A, not applicable; (A) = 30-day infection related mortality; (B) = 30-day all-cause mortality; COPD, Chronic obstructive Pulmonary Disease; HIV, Human immunodeficiency virus; HSCT, hematopoietic stem cell transplantation; TMP-SMX, sulfamethoxazole-trimethoprim.

## Data Availability

This study was registered with the trial registration number ID: CRD42020210843 under the international prospective register of systematic reviews (PROSPERO: www.crd.york.ac.uk/PROSPERO accessed on 24 November 2022). The datasets used and analyzed during the study are available from the corresponding author for all reasonable requests.

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
