# Peer review of "Combination Therapy versus Monotherapy in the Treatment of Stenotrophomonas maltophilia Infections: A Systematic Review and Meta-Analysis"

_antibiotics, 2022, doi:10.3390/antibiotics11121788_

Round 1
Reviewer 1 Report
The manuscript requires extensive English editing, to obtain a clear scientific message. I am afraid that having to infer the correct meaning of some sentences I reach the end of the subject without a clear understanding that the conclusions reported are based on the 16 studies reviewed or only the 4 studies included in the meta-analysis (I believe it is the latter, but I could've misinterpreted this).
Additionally, the instructions for authors should be followed. For instance, the abstract should not have more than 200 words.
Throughout the manuscript, the usage of italics for species should be ensured (see Introduction).
Table 1 could be reformatted and/or simplified (and a complete version added to the Supplementary Material) and it is currently unreadable. The outcomes of 'clinical response' or 'microbiological response' should be clarified. In Table 2, the first column is badly formatted as it is intended to be subsections for the remainder of the columns.
For Figure 1, why did the authors believe it is necessary to include the 'Previous studies' since both are n=0?
Author Response
Dear Reviewer,
On behalf of all the authors, I would like to resubmit our manuscript entitled: “Combination Therapy Versus Monotherapy in the Treatment of Stenotrophomonas maltophilia Infections: A Systematic Review and Meta-Analysis” to be considered for publication in your journal.
We have thoroughly revised our manuscript and supplementary file as well as responded point-by-point to each issue raised by the reviewers and editor as shown in this letter. We do appreciate all valuable comments from the reviewers and did our best to incorporate them in this revised version.
This paper has been published previously as a preprint at DOI: 10.21203/rs.3.rs-1212986/v1 and is not under consideration elsewhere. The authors are responsible for the reported research and have participated in the concept and design, analysis and interpretation of data, drafting or revising of the manuscript, and have approved the manuscript as submitted. The authors report no conflicts of interest.
Thank you for considering this paper for publication.
Sincerely,
Assistant Professor Sukrit Kanchanasurakit, PharmD

Reviewer 2 Report
Dear Authors,
The manuscript submitted by Abhisit Prawang et al. entitled `Combination Therapy Versus Monotherapy in the Treatment of Stenotrophomonas maltophilia Infections: A Systematic Review and Meta-Analysis` is attractive. Stenotrophomonas maltophilia is a prototype of bacteria intrinsically resistant to antibiotics, and this pathogen is frequently responsible for nosocomial outbreaks, especially in intensive care units. The therapeutic options regarding S maltophilia infections are minimal, owing to the intrinsic resistance of this pathogen to several classes of antibiotics.
First of all, I want to congratulate the authors for the idea of performing this meta-analysis.
The manuscript is well organized, especially the statistics performed are well presented. The English in some parts needs revision - l. 47-55, l.68-70, l.70-71, please reformulate; it is hard to understand.
1. To better justify the difficulty of treating S. maltophilia infections, the authors should include aspects related to the AMR of this pathogen in the Introduction and Discussions section from the following articles:
- PMCID: PMC6698998
- DOI: 10.1177/2333392819870774
doi.org/10.3390/antibiotics10101226
https://doi.org/10.3390/antibiotics11091263
doi:10.1177/2333392819870774
doi: 10.1080/14787210.2020.1730178.
2. l. 45-47, l. 53-56, l 66-67 needs references
3. Results section - the authors should try not to discuss the results in this section.
4. Table 1, please include the references in the column - author/year [x]
The following article should be included
PMCID: PMC8779545 DOI: 10.3390/pathogens11010081
5. l. 68-76 need a reference
6. The Discussion section needs to be revised; it is too short, only one paragraph.
7. l. 107-109 should be deleted. It is similar to l. 116
Author Response
Dear Reviewer,
On behalf of all the authors, I would like to resubmit our manuscript entitled: “Combination Therapy Versus Monotherapy in the Treatment of Stenotrophomonas maltophilia Infections: A Systematic Review and Meta-Analysis” to be considered for publication in your journal.
We have thoroughly revised our manuscript and supplementary file as well as responded point-by-point to each issue raised by the reviewers and editor as shown in this letter. We do appreciate all valuable comments from the reviewers and did our best to incorporate them in this revised version.
This paper has been published previously as a preprint at DOI: 10.21203/rs.3.rs-1212986/v1 and is not under consideration elsewhere. The authors are responsible for the reported research and have participated in the concept and design, analysis and interpretation of data, drafting or revising of the manuscript, and have approved the manuscript as submitted. The authors report no conflicts of interest.
Thank you for considering this paper for publication.
Sincerely,
Assistant Professor Sukrit Kanchanasurakit, PharmD
Responses to Reviewers
Manuscript ID: antibiotics-2009831
“Combination Therapy Versus Monotherapy in the Treatment of Stenotrophomonas maltophilia Infections: A Systematic Review and Meta-Analysis”
Reviewer 2
Comment #1
The manuscript is well organized, especially the statistics performed are well presented. The English in some parts needs revision - l. 47-55, l.68-70, l.70-71, please reformulate; it is hard to understand.
Response
Thank you for your valuable comment and recommendations. We agree with the reviewer suggestion. Thus, we have revised the English in manuscript by native English speaker.
Original (l. 47-55, l.68-70, l.70-71)
The main mechanism of resistance for this organism is the presence of its chromo-somes of genes that encode efflux pumps, as well as its antibiotic inactivating enzymes [2], and treatment for those who have a S. maltophilia infection, which is always resistant to first line therapy, is usually various combinations of antimicrobial agents. Regimens using antimicrobial agents have been surveyed extensively in order to achieve an efficient combination that will overcome bacterial resistance and attain synergism when possible. A combination of 2 to 3 conventional agents on S. maltophilia organisms that are particularly susceptible to these agents often lead to advantage our results, as are new antibiotics such as televancin that have demonstrated synergistic effects on S. maltophilia.
Moreover, in clinical practice should have considered between advantage of killing or stabilizing effect against bacteria and disadvantage of additive adverse events of drugs.
These studies still have been limited. Therefore, our purpose is to investigate the clinical outcome of combination therapy and the effect of monotherapy versus combination therapy for S. maltophilia infections and whether or not these very antithetical approaches affect mortality outcomes.
Revised We have revised all of manuscript by native English speaker in order to be readable. The details as attached manuscript file.
Comment #2
To better justify the difficulty of treating S. maltophilia infections, the authors should include aspects related to the AMR of this pathogen in the Introduction and Discussions section from the following articles:
PMCID: PMC6698998 DOI: 10.1177/2333392819870774
doi.org/10.3390/antibiotics10101226
https://doi.org/10.3390/antibiotics11091263
doi: 10.1080/14787210.2020.1730178.
Response
Thanks for your valuable comment and suggestions. The AMR is an important aspect in infectious diseases field. We have added these references into the manuscript according to your suggestions by added information about AMR of this pathogen in the Introduction and Discussions section.
References
Gajdács, M.; Urbán, E.; Prevalence and Antibiotic Resistance of Stenotrophomonas maltophilia in Respiratory Tract Samples: A 10-Year Epidemiological Snapshot. Health Serv Res Manag Epidemiol 2019, 15(6), 1-9.
Gibb, J.; Wong, D.W. Antimicrobial Treatment Strategies for Stenotrophomonas maltophilia: A Focus on Novel Therapies. Antibiotics 2021, 10, 1226.
Coppola, N.; Maraolo, A.E.; Onorato, L.; Scotto, R.; Calò, F.; Atripaldi, L.; Borrelli, A.; Corcione, A.; De Cristofaro, M.G.; Durante-Mangoni, E.; Filippelli, A.; Franci, G.; Galdo, M.; Guglielmi, G.; Pagliano, P.; Perrella, A.; Piazza, O.; Picardi, M.; Punzi, R.; Trama, U.; Gentile, I. Epidemiology, Mechanisms of Resistance and Treatment Algorithm for Infections Due to Carbapenem-Resistant Gram-Negative Bacteria: An Expert Panel Opinion. Antibiotics 2022, 11, 1263.
Petca, R.C.; Dănău, R.A.; Popescu, R.I.; Damian, D.; Mareș, C.; Petca, A.; Jinga, V. Xanthogranulomatous Pyelonephritis Caused by Stenotrophomonas maltophilia-The First Case Report and Brief Review. Pathogens 2022, 10, 11(1), 81.
Comment #3
- 45-47, l. 53-56, l 66-67 needs references.
Response
Thanks for your recommendations. We have added the reference to l. 45-47, l. 53-56, l. 66-67 already.
The reference of l. 45-47 is
Gil-Gil, T.; Martínez, J.L.; Blanco, P. Mechanisms of antimicrobial resistance in Stenotrophomonas maltophilia: a review of current knowledge. Expert Review of Anti-infective Therapy 2020, 18(4), 335-47.
The reference of l. 53-56 is
Hornsey, M.; Longshaw, C.; Phee, L.; Wareham, D.W.; In vitro activity of telavancin in combination with colistin versus Gram-negative bacterial pathogens. Antimicrob Agents Chemother 2012, 56(6), 3080-5.
The reference of l. 66-67 is
Shaik, S.; Amar, N.P.; Ramachandra, B.; Lalita, N. Case Reports Infective endocarditis caused by Stenotrophomonas maltophilia: A report of two cases and review of literature. Indian Heart Journal 2016, 68, s267 – s270.
Didier, P.; Valerie, F.; Jordi, M.; Jenneke, L.; Caren, B.; Pierre, T.;et al.Multicentric experience with interferon gamma therapy in sepsis induced immunosuppression. A case series. BMC Infectious Diseases 2019, 19, 931.
Iffat, K.; Aisha, I.; Farheen, A. Stenotrophomonas maltophilia Meningitis – A Case Series and Review of the Literature. Cureus 2020, 12(10), e11221.
Comment #4
Results section - the authors should try not to discuss the results in this section.
Response
Thanks for your valuable comment and recommendations. However, we have thoroughly read and have revised the manuscript by native English speaker. We could not find some discussion in the result section. Please feel free to let us know if there is some mistake for this issue.
Comment #5
Table 1, please include the references in the column - author/year (1) The following article should be included
PMCID: PMC8779545 DOI: 10.3390/pathogens11010081
Response
Thanks for your valuable comment and suggestions. We have reviewed and added this study and totally agree with reviewer recommendation and have revised the references in the manuscript as suggested. The following article be included in the manuscript Table 1.
Comment #6
- 68-76 need a reference.
Response
Thanks for your recommendations. We have added the reference to l. 68-76. Here are the original and revised versions.
The reference of l. 68-76 is
Hornsey, M.; Longshaw, C.; Phee, L.; Wareham, D.W.; In vitro activity of telavancin in combination with colistin versus Gram-negative bacterial pathogens. Antimicrob Agents Chemother 2012, 56(6), 3080-5.
Comment #7
The Discussion section needs to be revised; it is too short, only one paragraph.
Response
Thanks for your valuable comment and suggestions. We have revised the manuscript as suggested with the help of an academic native English speaker. The details as shown in Discuss section.
Comment #8
- 107-109 should be deleted. It is similar to l. 116
Response
Thanks for your valuable comment and suggestions. We have modified l. 107-109 as suggested by native English speaker to avoid similarity.

Round 2
Reviewer 1 Report
My concerns have been addressed.